# Between Silences: The Coronation of Portuguese Medieval Kings (12th–14th Centuries)

**Carla Varela Fernandes**

IEM-Instituto de Estudos Medievais, Universidade Nova de Lisboa, 1099-085 Lisboa, Portugal;
cfernandes@fcsh.unl.pt

**Abstract:** The coronations of Portugal's first dynasty constitute a complex topic. Approaching the theme requires understanding that an omission of words in written documentation can both affirm and deny possibilities. Likewise, visual documentation, such as illuminations, sculptures and other figurative arts, is scarce, raising a significant number of questions and thus is not trustworthy as a historical source. For this reason, the study of Portuguese coronations is filled with questions and silences. Art does not testify to these ceremonies, but shows that Portuguese kings valued regalia pertaining to both religious and secular ceremonies affirming their power, and that those insignias were different from those used by French or English kings in the same time period. In this study, I will use art, particularly funerary sculpture, but also objects with iconographic value, to demonstrate how these reflect elements of thought and the emotional pulsar of the various European societies that produced them.

**Keywords:** Portuguese kings; coronation; tomb sculpture; statues; *regaliae*; cathedral

---

## 1. A Brief Historiographical Summary on the Problematic of the Coronation of the First Portuguese Kings

For quite some time, authors were baffled by the lack of systematization of liturgical rites in the coronations of Portuguese and Castilian kings, particularly the act of anointing (Ruiz 1984; Linehan 1993; Rucquoi 1992). In the context of the deification of Royalty as claimed or even accepted in all Western European countries during the Early Middle Ages, such an absence became their focus, leading them to interpret medieval Iberian monarchies as anomalous.

Due to the lack of documentary or other information on intricate rites of enthronement, Portuguese historiography inevitably established parallels with other European kingdoms. While comparisons with France and England, for instance, reaffirmed the absence of such rites, the German tradition provided some similarities, notably the king's rise above a ceremonial shield by loudly cheering knights hailing him, a practice also observed in the Iberian kingdom of Navarra. Thus, the practice of liturgical consecration, or even of a conducive conjuncture enabling it, was excluded. Since only anointing conferred kings a divine character, such interpretations would inevitably cast doubt on the true sacredness of these monarchies.

However, subsequent studies on Castilian and Portuguese circumstances brought to the fore new approaches, as well as new evidence that the above-mentioned anomalies were neither straightforward in terms of their classification, nor that the French model took primacy over the affirmation of peninsular divine monarchies.

Regarding the Castilian case, Nieto Soria (1986, 1988) took a different approach to that of the preceding historiography, one that has been widely demonstrated in several studies.

Precedents for royal anointing at enthroning ceremonies as a form of alliance between the monarchy and the Church existed in Hispanic kingdoms and dated back to the Visigoth era. The anointing of

Afonso VII by the archbishop of Santiago de Compostela, described in *Historia Compostelana* (Book I, chp. 66), set one such precedent to the point where the continuity of this historical experience gave king Afonso X the assurance that, "*teniendo además en cuenta la más amplia dimensión autoritária que ahora se proponia extraer de la ideologia vinculada à monarquía de derecho divino no pareciese ni necesario ni coherente recurrir a tal recurso cerimonial*" [taking into account the proposed broader authoritarian dimension of the ideology of divine right connected to the monarchy, it would seem neither necessary nor consistent to resort to such a ceremonial resource] (Nieto Soria 1997, pp. 77–78). As far as Castilian monarchs were concerned, their power was legitimately divine for a long time, a stance/awareness that in the context of their time had the important theoretical support of both hierocratic and dualistic canonists (Kantorowicz 1985, pp. 84–85). In response to these arguments, Nieto Soria (1997) concludes that Afonso X saw anointing as a step back from the centralization and secularization of royal power, and not as a way to legitimate the divine origin that he took for granted anyway.

I believe that the same happened in Portugal, where the reigns of kings Afonso III, Dinis and even Afonso IV were profoundly impacted by the influence of the Castilian court, particularly in the case of Castilian King Afonso X (who influenced both family relations and the continuous progression in the centralization of regal power), but also by the political personas of Sancho IV and Afonso XI.

The consecration and coronation of Portuguese kings and their wives has a long historiography. I am mainly interested in highlighting the various positions of different authors from the timeline of their writings on the issue.

Brásio (1962, pp. 21–49) and Merêa (1962, pp. 6–11) categorically stated that, despite King Duarte's many requests to the pope, Portuguese kings were never crowned or liturgically consecrated. According to these authors, enthroning consisted of a ceremony where knights lifted Portuguese monarchs above a shield while hailing them as new kings. Their conclusions were based on the absence of descriptions in *The Chronicles* of Fernão Lopes and Rui de Pina. When, confronted with Friar António Brandão's description of the anointing and coronation of King Sancho I at Coimbra's Cathedral, Paulo Merêa casted it aside as an unreliable source. Both Brásio and Merêa followed the reasoning of Spanish historians, particularly that of Sanchez Albornoz (1962, pp. 5–36), but also Schramm's (1960) classic research, which were biased opponents of considering the existence or at least the continuity of consecration and coronation ceremonies in Hispanic kingdoms (Mattoso 1993).

The questions Albornoz and Schramm raised both led to the emergence of a new school of thought in Spain and triggered many doubts that became the basis of Teófilo Ruiz' thesis in 1984. While in Spain, Nieto Soria took on the task of thoroughly reviewing, analyzing and updating the content, in Portugal it was up to José Mattoso to re-examine, not just the Portuguese case, but also the same Castilian and Leonese sources that Albornoz and, particularly, Teófilo Ruíz had based their writings on (Mattoso 1993).

Refuting hypotheses based on the terseness of peninsular chronicle sources regarding the matters of consecration and coronation, Mattoso pondered on some rather expressive excerpts, noting the presence of elements pertaining to the sacred. Namely, Mattoso considered references to the clergy's presence in coronation ceremonies and the temples in which these took place, as mentioned in both the *Crónica Latina de los Reyes de Castilla* or the *Primera Crónica General de España*. An example is the well-known coronation of Afonso XI, which took place in a church (Mattoso 1993, pp. 189–190).

Even though early Portuguese chronicles were deeply laconic on this issue, *Crónica de 1419* includes the most significant sentence in reference to king Sancho I: "foi coroado por rei em Coimbra" [crowned as king in Coimbra], a more direct description than the usual references to "alçaram por rei" or "foi alçado por rei" [risen as king], found in *Crónica de 1344*, as well as in passages of Rui Pina's chronicles of the 15th century (Pina 1945). According to him, when King Dinis' father died, he "*foy logo aleuãntado e obedeçido por Rey de portugal E do algarue ( … )*" [was immediately risen and obeyed as King of Portugal and The Algarve]. King Afonso IV was also "*solenemente alevantado e obedecido por rei* [solemnly raised and hailed as king] according to the author of *Crónica de 1419*. Fernão Lopes, author of *Crónica de D. Fernando* mentions the Monastery of Alcobaça, an important ecclesiastical space, as the

site where the king rose to power in the presence of a priest: *"el d'aquel moesteiro [Alcobaça] onde seu padre fora tragido e el levantado por rei [ . . . ]"* (Mattoso 1993, p. 190).

José Mattoso interpreted the chronicler's brief descriptions as a lack of interest in regal investiture, or simply, as an inability to put any sign language into words, regardless of how solemn this might have been. However, Mattoso found it premature to state that Portuguese monarchs were deprived of any consecration and coronation ceremonies, simply from these sources. In fact, during the first dynasty, several accounts appear to demonstrate otherwise.

Following Mattoso's own enumeration, such accounts are provided in *Manuscripto 1134* from the Municipal Library of Porto, dated from the 12th century and known as *Pontifical de Santa Cruz de Coimbra*, which contains the *Ordo Benedicendi Regnum* in Folios 130 to 134. Beyond the evidence of its use, it includes a prayer added to the margins, introducing a special solemnization blessing. The *Ordo* also mentions the clergyman's anointing of hands, chest, back and arms, the blessing and offering of the sword, the bracelets, the solemn cape or *pallium,* the staff and the crowning. In addition, the investiture of these insignias took place before mass. In comparison to other western *Ordines*, the document is copied from the ritual described in *Pontifical Romano-Germano do Século X*, the most disseminated document among Roman liturgy, and of which there is another 12th century well-known copy, with some variants, in the so-called *Ceremonial Cardeña*. "All this shows how widespread the *Ordo* was, probably since after the abandonment of Hispanic Liturgy at the end of the 11th century" (Mattoso 1993, p. 192). Still, according to Mattoso, it is probable that the manuscript was used for the coronation of King Sancho I and the following monarchs.

Also kept in the Municipal Library of Porto and also originated in Coimbra, *Manuscripto 343* describes a simplified ritual with some variants from the precious *Ordo.* Its usage or practice cannot entirely be denied or cast aside.

A third testimony comes from the coronation ceremony of Afonso XI and that of his son Fernando V. Mattoso finds the description relevant to the Portuguese case, since it was authored by Raimundo Ebrard II, Bishop of Coimbra, sometime between 1325 and 1333. Written in both Castilian and Latin, it denotes many *portuguesisms* (features peculiar to the Portuguese language), previously underlined by its editor, Sánchez Albornoz. It is a testimony to the interest the Portuguese church took in the liturgical solemnity of the coronation act. The document "... also includes the anointing of the shoulders and back of both King and Queen, the blessing and offering of the sword and crown, as well as the enthronement on a podium placed inside the church" (Mattoso 1993, p. 192).

The fourth and last testimony, which is not as apparent but is clear enough, is included in *Livro dos Arautos*, dated from 1416 and published in 1977 by Aires de Nascimento (pp. 250–251). A Portuguese herald (*arauto*), who travelled around various European courts, referred to Coimbra's Cathedral as the site of coronations *ex consutudine* of Portuguese kings. "Now, if it was a cathedral and a coronation, it must be presumed that the ecclesiastical authorities approved the use of liturgical ceremonies. And if it was customary, not just one but several kings were subjected to it" (Mattoso 1993, p. 193).

J. Mattoso concludes that the anomaly verified by chroniclers' lack of information on the religious consecration ceremony is due to the fact that "the court was not keen on emphasizing the many acts and expressions of ecclesiastical submission to the king pertaining to these rituals, and described in both the Roman-German Pontifical and Bishop Raimundo Ebrard's account" (Mattoso 1993, p. 198). As mentioned before, a conclusion close to that of Manuel Nieto Soria in regard to Castilian sources.

In view of these well-sourced arguments, it is clear the consecration and coronation ceremonies took place in Portugal, at least during the early reigns. It is possible that not all monarchs in the first dynasty were anointed and surely some took for granted their own divinity based on their ancestors' previous consecrations. In doing so, they avoided subjecting their own power to ecclesiastical authorities, an analogous situation to that of Castile. Early monarchs however, as supported by J. Mattoso himself based on Raimundo Ebrard's testimony, would not dismiss the act of anointing as a recognition of their own power by another greater and superior one. According to Ebrard, kings *"se vivem a seruicio de Dios, faran milagros en sus vidas"* [if they live to the service of God, they will perform

miracles]. This conferred monarchs with an ability to perform miracles and make justice: "*que los reyes que quieren guardar iusticia, solamiente com los ojos destruen todo mal*" [that kings wanting justice to prevail, can destroy evil with their look alone] (Mattoso 1993, pp. 198–99).

Although this is a plausible reality from the beginning of Sancho I's reign on, caution is advised when it comes to the first Portuguese king, Afonso Henriques. This should be the case even while considering that his ceremonial coronation resulted from the political maneuvers of notable ecclesiastics who advised him while he was still a prince, since territorial characteristics and military victories were the real basis of his leadership and legitimacy in ascending to the throne of the newborn kingdom.

A few years ago, a study by A. F. Pimentel brought about new and interesting observations on the coronation and consecration of early Portuguese monarchs. These pointers departed, not from the discovery of new documents or chronicles, but from an iconological analysis of the spatiality and foundation elements of Coimbra's cathedral (plan and elevation). The building dates from the 12th century and is situated in the kingdom's first court, which comprised both the royal palace and religious quarters. Monarchs gave support to ecclesiastical authorities and, in turn, the clergy provided kings with both spoken and written legal services. Documents were produced by highly prestigious clergymen from both the Santa Cruz Monastery (functioning both as an archive of regal documents and as royal treasure) and the Cathedral where bishop and priests would have been responsible for organizing the religious aspects of enthronement rituals at least since King Sancho I (r. 1185–1211), and perhaps even during the reign of his father (Afonso Henriques). According to Pimentel, the Cathedral's architecture largely responded to different parts of the ceremony; the architecture responded to both the private instances reserved for peer acclaim and the public moments aimed at popular cheering (Pimentel 2004, pp. 87–122). Although the space provided here to expand on these ideas is limited, in my view their proposition makes a very strong point. They disrupt the early silence of sources, upon which the early historiography drew its conclusions, and adds to the buzz caused by José Mattoso and other historians' own deviation from those previous beliefs. The iconography itself, neither abundant nor artistically relevant (albeit truly exceptional for the case in point), sheds some light on an initially shadowy topic (Fernandes 2004).

## 2. What 12th to 14th Centuries' Iconography Tells Us?

Unlike France or England, in Portugal there is nothing compared to a consecration *Ordo* with illumination drawings or paintings, which were so common in the Early Middle Ages; there is nothing like Bayeaux tapestry (which in fact is a large-scale embroidery, but is known to history as a tapestry) or even wax seals and minted coins of great significance. Among all the objects composing the regalia of the first reigns, these are described as offers made during the consecration ceremonies, at least for the second king, Sancho I, as cited in the *Pontifical de Santa Cruz de Coimbra*. Regardless of their artistic value, and whether they are recorded as having existed and are now gone, they must be appreciated as forms of representation contemporary to the monarchs.

There is some consensus in the historiography of art regarding two stone sculptures placed in two different cities as being representations of the first monarch of Portugal, King Afonso Henriques, whose reign lasted from 1139 to 1185 (Fernandes 2004, pp. 572–82 and Fernandes 2017, pp. 77–90). The first and perhaps oldest (there is no information stating the exact dates in which each sculpture was produced) is part of a set of two, which was naively designed and cut, and carved out of granite in a rather coarse manner. The second sculpture in this set, which is aesthetically similar and certainly sculpted by the same craftsman, represents a bishop, probably João Peculiar, a strong supporter of the recognition of Portugal as an independent kingdom and of Afonso Henriques as its first king. Both sculptures were destined to the church of the old Benedictine monastery of S. Pedro de Rates.

As a visual document, the iconographic elements in the king's sculpture are of great relevance and importance. The king is represented wearing a crown, his body covered by a long garment, wielding a sword over his shoulder. The dimensions of both sword and crown are indicative of their prominence among all the objects composing the early monarchs' regalia, at a time of territorial reconquest in

the Peninsula. The king was an entitled sovereign by the sword he held and which defined him in his condition of crusader and conqueror of lands for the Christians. Later, his worth was recognized both by Rome and the emperor, his cousin King Afonso VII of Leon and Castile (r. 1126–1157), since opposing views did not carry the same political and religious weight to the Pope (Figure 1).

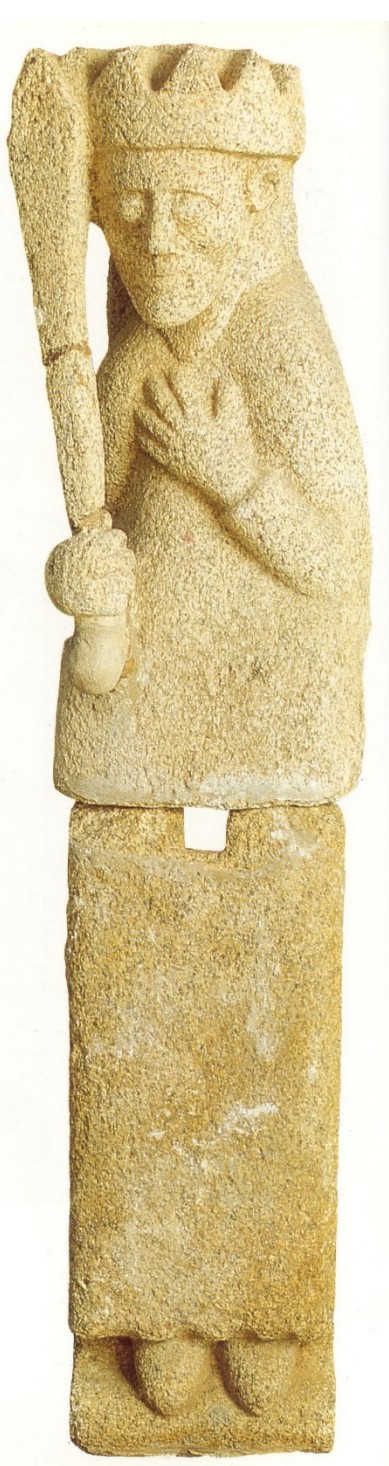

**Figure 1.** Afonso Henriques (?). 12th century. Church of the monastery of São Pedro de Rates.

The second statue, presumed to represent the same monarch, is incomplete, broken in two at some point in time, and all that is left is the upper part of the body. Interestingly, the most valued regal attributes are still the crown as an essential element, the sword which is also of great size and placed

over the shoulder as well as what was probably a scepter, shaped at one extreme as a Latin cross and held by the king on the side. The manner in which a string of beads fastens the sword's sheath is unprecedented. As for the cross and from the way that the hand holds it, there is no indication that it was attached to an orb, or any signs that a globe even existed. However, the hand's position offers the possibility for the existence of some sort of rod that may have disappeared with the lower part of the body. The eyes, the worn out beard and the hair sculpted in grey marble are all consistent with western figurative art from the 12th and 13th centuries. All of these features can be representative of both Afonso Henriques or of his son Sancho I. The sculpture was initially placed at the Church of Santa Maria da Alcáçova, and was perhaps moved to a door niche at the same place before being taken to the Archaeological Museum of Carmo in Lisbon, according to the museum's old inventory (Fernandes 2005, p. 342; Fernandes 2017, pp. 77–83) (Figure 2).

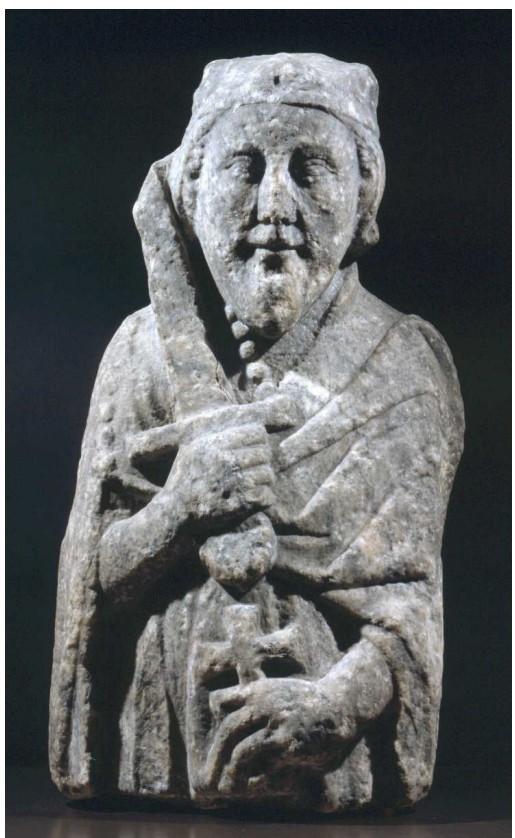

**Figure 2.** Afonso Henriques (?). 12th or 13th century. From Santarém, currently at the Archaeological Museum of Carmo (Lisbon). Photo: ©José Pessoa/ADF-DGPC.

With regard to the imagery extolling empowerment and the exercise of monarchic power, it is important to value other types of objects. For some kings, whose military action and victories in battles were decisive for the conquest of territory, the definition of borders or, simply, for the pompous demonstration of the Christian mission they were invested in, the sword, the shield and victory banners all possessed a high symbolic power and were collectively recognized. These were all reasons for placing these objects in very visible spots, close to their memorials and grave sites.

The king's sword and shield were displayed next to the tomb of Afonso Henriques on the porch of the Monastery of Santa Cruz de Coimbra. The sword remained there at least until the 16th century when King Sebastião requested it from the Canons Regular of Saint Augustine to take to the Battle of Alcácer-Quibir (where it was lost along with the king). The shield, placed over the king's tomb, symbolized the one used at the Battle of Ourique, as well as that over which the king had been acclaimed. It disappeared in the 12th century after the tomb as it was considered too modest by king

Manuel I, and was replaced by a much different one (Mattoso 2006, pp. 120–121, from previous sources). The *Livro dos Arautos* narrates some notable and enduring legends that honor the symbolic value of these two objects, belonging to a then recent and warrior-like monarchy (ed. A.A. Nascimento 1977, p. 251).

The founding king's material shield was thus venerated as a kind of relic, with almost magical power attributes. ( . . . ) The shield was a physical testimony to the supernatural protection and divine sanctioning pertaining to the royal dignity of its bearer, also functioning as a reminder of the victory at *Ourique* and the acclamation of the Prince as King at the battle's site. (Mattoso 2006, pp. 221–122)

The "magical aura" attributed to Afonso Henriques' weapons was revived two centuries later by King Afonso IV (r. 1325–1357). Based on what I could gather and reflect upon in regard to this hypothesis, I understand that there was an ideological and political intention in associating Afonso IV's image to that of the founding and mythical first monarch (Fernandes 2004; Fernandes 2007, pp. 163–16). This was due to both the importance and valorization given to the relics of Saint Vincent (kept in the Lisbon Cathedral where he was buried—*ad sanctus*—and brought to the city in the political context of Afonso Henriques' government in 1173—(Gomes and Nascimento 1988), and the monarch's order to display looted objects from the Battle of Salado (fought against the moors in 1340 in partnership with his son-in-law Afonso XI of Castile) along the Cathedral's walls and later close to his tomb (da Fonseca 1728, p. 60; de Sousa [1739] 1946–1954, p. 307). Among those mythical objects, none made its way to us and all that remains are memories recorded in various writings, especially chronicles, written long after the events took place.

Within the scope of iconography validating the warrior character of early kings, it is interesting to observe two ink drawings that represent two older pendant seals, as a means to record previous stamp elements when they came into disuse (copy from *Foral de Penas Roias do Livro de Doações de Afonso III*, IANTT, Gavetas, gav. 15, mç. 10, doc. 14 and *Livro das Doações de Afonso III*, fol. 13v,o). The drawings were produced in the second half of the 13th century and in the 14th century, respectively, and reproduce both the shape of the stamps and the king's effigy consisting either of a single seal matrix or two similar ones. Both drawings represent the effigy of King Sancho I (r. 1185–1211) in majesty (Branco 2005, p. 162). The authors of the respective documents in which we find these equally rudimentary and expressive representations were almost certainly also the artists of the drawings, although their concerns were less artistic than iconographic. They were interested in accurately representing the figure of the king in the old seals, with his regal coronation accoutrements. The king is portrayed in his solemn attire, one arm raised with his hand gesture suggesting the offering of a blessing to his subjects (Branco 2005, p. 162). The crowns in both drawings are faithful copies of the originals: opened and topped, not by *fleurs-de-lis*, but rather by the Greek Cross, as it was used in Portuguese metalworking during the 12th and early 13th centuries. These are the only representations of a first dynasty king displaying his sword, spurs (both symbols of his warrior-like character) and banner (Fernandes 2005, p. 584). In the 14th century representation, the banner is shown suspended on air in the left flank, while the 13th century drawing displays it as being grounded. To sum up, the crown remained a privileged insignia of royalty, but the sword was replaced by the banner; this is an image that evidences the warrior nature of early kings as military leaders and guardians of important victories in the battlefield (Figures 3 and 4).

These and other examples show that, like other kingdoms, Portugal revived the paradigm of a "new David" or a "new Charlemagne"; model images of great and mythical knights incorporated into Portuguese kings. However, Portuguese monarchy did not just follow its contemporary governmental proposals; instead, the monarchy created its own legends, including the myth of origins and that of the ideal monarch. However, these legends were inspired by preceding models, as I believe was the case for the two monarchs mentioned above.

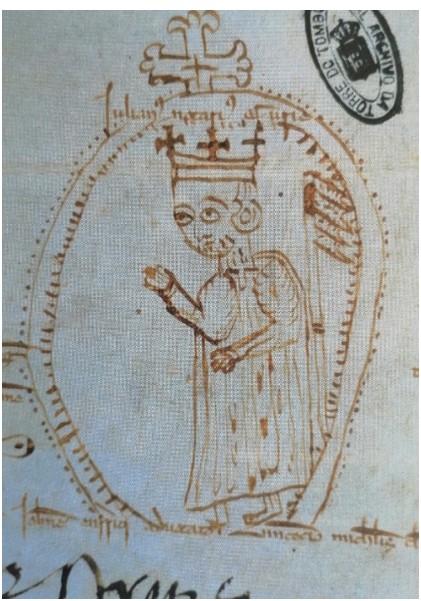

**Figure 3.** Sancho I. Drawing of the royal seal on a document (ANTT, Gavetas, gav. 15, mç. 10, doc. 14). Second half of the 13th century. Photo: ©ANTT

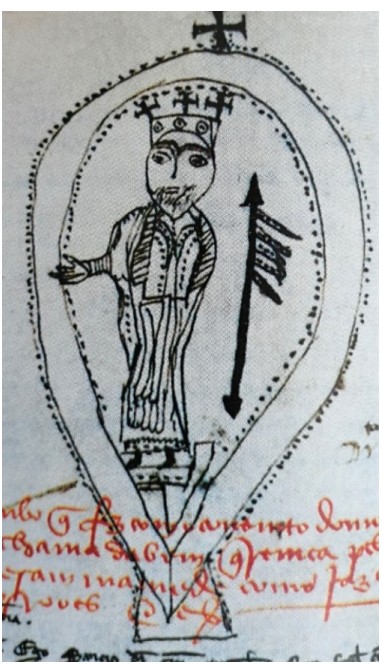

**Figure 4.** Sancho I. Drawing of the royal seal on a document from the Donations Book of Afonso III, fol. 13v°. 14th century. Photo: ©ANTT

A case in point is that of King Dinis (r. 1279–1325), known, among other aspects, for his interest in culture, particularly *troubadour* culture, and for the fact that he himself was a poet of songs. His personality is compared to that of his grandfather, Castilian King Afonso X, known as The Wise, following a model with biblical roots, but one also also found among Iberian late medieval monarchs. However, art has not left us much iconography or artistic expression allusive of such a monarchic ideal (as did France some years later during the reign of Charles V). When analyzing the iconography that King Dinis himself chose to adorn his tomb (placed in the Cistercian Monastery of São Dinis e São Bernardo de Odivelas, and severely mutilated after the collapse of the dome during the 1755 Earthquake), the importance becomes clear of being represented in full coronation regalia (crown and

garment), both in the high relief sculpture and during his last actions preparing for the so-called *Good Death*, as a Christian king.

The sword, buried with the king, can no longer be found under the sarcophagus' cover, but the spurs carved in the horizontal high relief sculpture are still visible. Both sword and spurs, the two insignias indicating knighthood, were common in royal tomb sculptures, particularly this entire dynasty, whose affirmation of a warrior-like character originated in the first king (even though his tomb did not have a sculpture; instead, the tomb only had the sword and shield displayed next to it).

The tomb statue of Afonso IV (also destroyed during the 1755 Earthquake) is unknown. However, judging by the tomb of his first knight, Lopo Fernandes Pacheco (another victor of the Battle of Salado, also buried in Lisbon's Cathedral), as well as by that of his son and future king, Pedro I (r. 1357–1367), situated in the Monastery of Alcobaça, he must have been proudly represented with his sword and spurs, as well as his banner and other regalia displayed on the walls near the tomb.

Pedro I was king for 10 years only, but he took good care of his social image, although there is not any written or visual documentation left to inform us about his coronation ceremony. This occurred for certain, but apart from the crown and the sword, the king's iconographical record is limited to his sarcophagus, where he is presented as knight-king in an imposing statue, dressed in a long and solemn garment, holding the sword sheath with both hands and wearing a crown apparently sculpted after actual gold smithery models. In the scene at the top of the sarcophagus, where he is represented on the upper axis of the *Wheel of Life* as an enthroned king, only the solemn garment is visible. Neither his crown nor the objects he held in his hands survived due to the advanced state of deterioration of the stone (Figure 5).

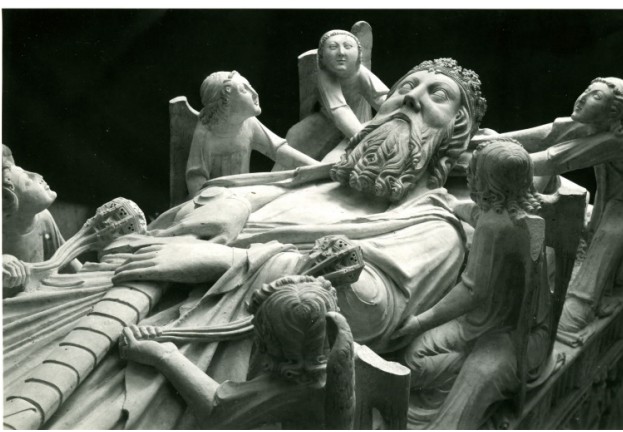

**Figure 5.** Detail of the lying statue of King Pedro I. Ca. 1357–1361. Alcobaça Monastery Church. Photo: Mário Novais (CRSIQS).

The last monarch in the Burgundy dynasty, Fernando I, was crowned in the Royal Abbey of Santa Maria de Alcobaça, a sacred space of great religious, social and political relevance. There, the tomb of his father, awaiting the body of the king perished in Estremoz a couple of days before, faced that of his step-mother, Inês de Castro, murdered in 1335 and entombed there in 1361. Although Fernando was acclaimed there as King of Portugal and The Algarve on 20 January, 1367, he chose to be buried elsewhere, breaking the tradition of using this Abbey as the first dynasty's pantheon (he was not the only one to break this tradition though). Instead, he prepared a monumental tomb for himself and his mother's remains. She was the daughter of an important Castilian nobleman and in Fernando's mind, Queen and the sole legitimate wife of King Pedro I. His tomb was the only Portuguese royal tomb from the 14th century that does not have a statue. The king is represented in a small bust, which is crowned and displayed on the same symmetrical axis as the bust of Christ of identical size. Its iconography is varied, complex and intriguing, offering many clues as to how this King saw his role as a dignified

monarch and his main concerns in the face of death (Fernandes 2009). However, it sheds little light as to the aspects of his own coronation ceremony (Figure 6).

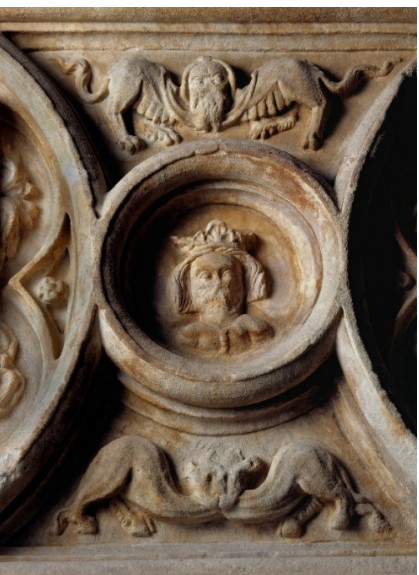

**Figure 6.** Bust of King Fernando I on the cover of his tomb. 1380–1383. From Santarem, currently at the Archaeological Museum of Carmo (Lisbon). Photo: ©José Pessoa/ADF-DGPC.

Lastly, I would like to point out that a 14th century Castilian manuscript (1312–1325) entitled *Compendio de Crónicas de Reyes* (*Biblioteca Nacional de España*) contains representations of first dynasty Portuguese kings and, in spite of the idealization, even standardization of these portraits, they provide consistent evidence of the regalia associated with the rituals of coronation and the insignias offered to the kings in those ceremonies. These representations have essentially been used to illustrate online biographies of the monarchs, but they should in fact be valued as idealized figurations of the kings that preceded the times of King Dinis, when they were produced. The images were not part of a Portuguese artistic commission, and cannot be taken at face value (no representation of our kings previous to the 15th century can), but they illustrate the ways in which kings were imagined or remembered before 1325 (Figure 7).

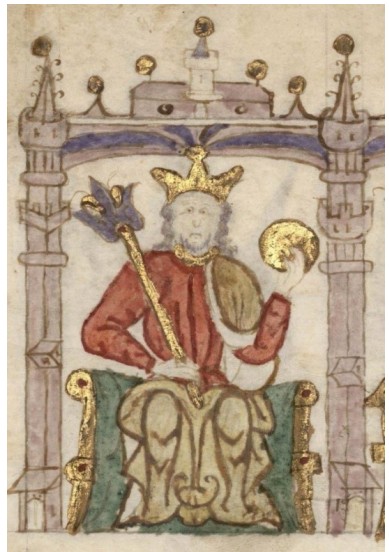

**Figure 7.** D. Sancho II—Compendio de crónicas de reyes (Biblioteca Nacional de España), 1312–1325.

## 3. Final Remarks

Among the statues of Afonso Henriques, the copies of Sancho I's seals, the tombs of monarchs sculpted with scenes of their lives and the tomb covers displaying sculptures, the iconography is not more eloquent than the existing documentation in describing the process of coronation rituals and its valued insignias between the 12th and 14th centuries. Surely, the constant presence of offensive or defensive weapons that these monarchs are represented with make it clear that the emphasis is on the warrior character of the various monarchs, and that is obviously significant. This feature is also something that brings closer the various iconographies of peninsular monarchs from various kingdoms during the centuries marked by territorial conquest by Muslims. If we look at a legendary/literary tradition associated with the lives of some kings or military heroes, from Antiquity to the Middle Ages (it is enough that we remember the famous and named swords of Alexander the Great, or King Arthur, for example), so that it is not unreasonable to think of the constant and so significant presence of the sword as an element with a "divine" character and to associate that element with the divine origin of the royal power of the peninsular kings, namely the Portuguese kings and the sacrality of the royal authority (this is just another question that remains open, but on which it will be worth reflecting).

In regard to the consecration rituals of the monarchs' respective queens, documentation is entirely absent and the iconography, be it in tombs or on seals, is limited to the use of the crown as a main regal insignia. All other attributes make reference to their virtues and some aspects of their biography.

We learnt, mainly through the research of José Mattoso and Aires do Nascimento, that the coronation ceremonies of Portuguese kings followed principles and customs rooted in Visigoth monarchies (similarly, Hispanic or Mozarab liturgical rites from those times survived for a long time). However, they also included customs later introduced and adapted in western Christian kingdoms, particularly in France where manuscripts specific to these ceremonies were produced in large quantities. Some of these manuscripts were very useful and clear, containing numerous illuminations that visually translated the consecration words in the *Ordos.* In Portugal, manuscripts where images played such a crucial role were either non-existent or have entirely disappeared with the passing of time. Thus, in addition to the previously mentioned texts, we value any bit of information that material culture can provide, in order to help us understand a reality pertaining to the ecclesiastical spaces in the kingdom (cathedrals and monasteries), between the 12th/early 13th centuries and the late 14th centuries. These are parts of material culture that are, however, deeply marked by a chiaroscuro of absences.

**Funding:** This research was funded by Fundação para Ciência e a Tecnologia/Ministèrio da Ciência e Tecnologia (2012–2017).

**Conflicts of Interest:** The author declares no conflict of interest.

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
