# Peer review of "Between Silences: The Coronation of Portuguese Medieval Kings (12th–14th Centuries)"

_arts, 1932_

Round 1

Reviewer 1 Report

line 208: the church is Santa Maria da Alcáçova and not São Miguel da Alcáçova

line 335: missing title of the picture

Author Response

The two sugested have already been introduced in the revision draft

Thank you for de reading and reviewing.

Reviewer 2 Report

The text is a review of historiography on the subject of coronation ceremonies of the Kings of Portugal (12th-14th centuries). The author clearly states the state of arts and the objectives of his paper. The novelty of the article is the incorporation of the iconographic approach to enrich the scarce knowledge that written sources provide about coronation ceremonies in this period in Portugal.

Author Response

The two suggested changes have already been introduced in the revision draft.

Thank you for de reading and reviewing. 

Reviewer 3 Report

The paper deals with highly interesting topic and it is very important that in this way some of the most interesting points of Portuguese medieval history are disponible to everyone.

Because of that I could suggest just some clarifications. I think it will be very interesting to have a historical introduction on the events that brought to first coronations. Besides that, it would be of highest interest to make paralells with primary sources and their data on coronations and to make compariosons with iconography. It seems that there is too much historiography and too little primary sources, and I think it would be much more to provide also data by the primary sources in order o clarify the coronation itself. The author gave an extensive insight into the Portuguese historiography, which is very valuable for wider audience. What could also be useful is a brief insight on the Portuguese medieval heraldry and visual representations, just to help the reader.

After these minor changes, and some additions, I fully recommend publishing this interesting paper.
